# An Integrative Study Showing the Adaptation to Sub-Optimal Growth Conditions of Natural Populations of *Arabidopsis thaliana*: A Focus on Cell Wall Changes

**DOI:** 10.3390/cells9102249

**Published:** 2020-10-07

**Authors:** Harold Duruflé, Philippe Ranocha, Thierry Balliau, Michel Zivy, Cécile Albenne, Vincent Burlat, Sébastien Déjean, Elisabeth Jamet, Christophe Dunand

**Affiliations:** 1Laboratoire de Recherche en Sciences Végétales, Université de Toulouse, CNRS, UPS, 24 chemin de Borde Rouge, 31320 Auzeville-Tolosane, France; harold.durufle@inrae.fr (H.D.); ranocha@lrsv.ups-tlse.fr (P.R.); cecile.albenne@ibcg.biotoul.fr (C.A.); burlat@lrsv.ups-tlse.fr (V.B.); jamet@lrsv.ups-tlse.fr (E.J.); 2PAPPSO, Université Paris-Saclay, INRAE, CNRS, AgroParisTech, GQE-Le Moulon, 91190 Gif-sur-Yvette, France; thierry.balliau@inrae.fr (T.B.); michel.zivy@inrae.fr (M.Z.); 3Institut de Mathématiques de Toulouse, Université de Toulouse, CNRS, UPS, UMR 5219, 31062 Toulouse, France

**Keywords:** *Arabidopsis thaliana*, cell wall, integrative study, natural variation, temperature adaptation

## Abstract

In the global warming context, plant adaptation occurs, but the underlying molecular mechanisms are poorly described. Studying natural variation of the model plant *Arabidopsis*
*thaliana* adapted to various environments along an altitudinal gradient should contribute to the identification of new traits related to adaptation to contrasted growth conditions. The study was focused on the cell wall (CW) which plays major roles in the response to environmental changes. Rosettes and floral stems of four newly-described populations collected at different altitudinal levels in the Pyrenees Mountains were studied in laboratory conditions at two growth temperatures (22 vs. 15 °C) and compared to the well-described Col ecotype. Multi-omic analyses combining phenomics, metabolomics, CW proteomics, and transcriptomics were carried out to perform an integrative study to understand the mechanisms of plant adaptation to contrasted growth temperature. Different developmental responses of rosettes and floral stems were observed, especially at the CW level. In addition, specific population responses are shown in relation with their environment and their genetics. Candidate genes or proteins playing roles in the CW dynamics were identified and will deserve functional validation. Using a powerful framework of data integration has led to conclusions that could not have been reached using standard statistical approaches.

## 1. Introduction

In the global warming context, temperature fluctuations will be major concerns [1]. Local adaptation is defined as the response of natural populations following their interactions with their environmental conditions. Freezing events, without any preceding chilling period, associated to sudden elevations of temperature, are critical for plant development. Thus, coping with thermal constraints will be a major challenge to maintain agricultural productivity by doing a selection of warm-adapted and cold-resistant species [2]. Looking at natural variations within wild species is crucial to elucidating the molecular bases of the phenotypic adaptation [3].

Natural abiotic gradients, such as mountains, provide a powerful setting to study how plant species adapt to contrasted environmental factors [4]. For example, with the elevation of the altitude in temperate climates, plants endure important environmental variations such as temperature diminution and humidity increase, combined to the rise of UV radiations [5]. In heterogeneous habitats, plants are adapted to the prevailing conditions [6,7,8], and/or respond to contrasted environmental conditions thanks to their phenotypic plasticity [9,10]. Several studies have shown that the plant phenotypes such as biomass, height, number of leaves, flowering phenology or reproductive rate can change along an environmental gradient [11,12,13]. However, the molecular mechanisms associated are still poorly described [14,15,16].

Cell walls (CWs) represent a dynamic extracellular compartment that contributes to modify the cell and plant shapes at any time during development [17]. The so-called primary cell walls are mainly composed of polysaccharides (cellulose, pectins and hemicelluloses) and of proteins. The secondary walls are synthesized at the end of growth and can contain lignins. The cell wall composition and structure can vary upon changes in developmental and environmental conditions (for a review, see [18]). As an example, a cold stress can induce an increase in pectin content [19]. The cuticle has also been shown to play critical roles during development and in response to biotic and abiotic stresses [20,21,22,23]. CW proteins (CWPs), that drive the remodeling of CW polysaccharides and mediate cell-to-cell signaling, are essential for the control of growth and CW integrity, as well as for the response to stresses [24,25,26].

The well-known model plant *Arabidopsis thaliana* (L.) Heynh (*Brassicaceae*) is a self-fertilizing and annual species with a worldwide distribution. Several researches about the natural genetic and phenotypic variation of *A. thaliana* along altitudinal gradients have been published [27,28,29,30,31,32,33,34]. These studies have highlighted the importance to combine multi-level datasets (e.g., environmental, genetics, morphological). Taken together, the results show strong links between the spatial distribution of plants and the genetic diversity, potentially reflecting adaptation to low or high altitude. The phenotypic differentiation may result from interactions between adaptation to local microhabitat and the demographic history influenced by glaciation cycles or by the recent seed dispersal [35]. *A. thaliana* capacity to adapt to diverse environmental conditions make it a very appropriate model for phenotypic plasticity studies [16,36,37].

*A. thaliana* natural diversity is a major resource to study local adaptation to major temperature variation. In order to evaluate the genetic background contribution to CW plasticity at sub-optimal temperature growth conditions, the well-described Col ecotype of *A. thaliana* as well as four newly-described populations originated from contrasted natural environments in the French Pyrenees Mountains [38] were studied. Two vegetative organs were analyzed, the rosette and the floral stem. Two different growth temperatures (22 and 15 °C) were used in laboratory conditions to simulate the temperature effect. The 22 °C temperature has been chosen as the optimal growth temperature for the Col ecotype, whereas the 15 °C temperature was used to simulate the higher altitude temperature conditions [38]. To tackle the complex mechanisms of adaptation to sub-optimal temperatures, a systems biology approach combining climate parameters, phenomics, metabolomics, proteomics and transcriptomics has been developed. The statistical integrative analysis of the various heterogeneous omic datasets using the mixOmics package [39,40] allowed identifying new genes/proteins possibly involved in the phenotypic plasticity responses of the four Pyrenean populations and the Columbia ecotype (Col) as a reference to temperature changes, with a specific focus on the CW.

## 2. Materials and Methods

### 2.1. Plant Material

The annual plant *A. thaliana* was studied: the Columbia ecotype (Col) originating from Poland where it was originally collecting at 200 m in altitude, as a reference; and four Pyrenean populations named after their geographical origin Roch (Chapelle Saint Roch), Grip (Grip), Hern (Herran) and Hosp (L’Hospitalet-près-l’Andorre), which are living between 700 and 1400 m in altitude in the Pyrenees Mountain [38]. We have used one genotype per homogeneous population, and 20 plants of this genotype for each of the three biological replicate for all the analyses. Seeds were sowed in Jiffy-7^®^ peat pellets (Jiffy International, Kristiansand, Norway). After 48 h of stratification at 4 °C in darkness, plants were grown at two different temperatures, at 22 or 15 °C, under a light intensity of 90 µmol photons.m^−2^s^−1^. They were cultivated under a long-day condition (16 h light/8 h dark) with 70% humidity. Rosettes were collected just before bolting (stage 5.10 [41]) from four- or six-week-old plants grown at 22 or 15 °C, respectively. Floral stems were also collected at the same stage of development (first flower, stage 6 [40]): 6 weeks for Col; 7 weeks for Grip and Roch; and 8 weeks for Hosp and Hern. They were collected two weeks later for all the plants grown at 15 °C.

### 2.2. Macrophenotyping

Rosettes of plants grown at 22 and 15 °C were analyzed at the time of sampling. We have measured their diameter and fresh mass. The number of leaves was counted. Before freezing, pictures were taken to measure the rosette areas with the ImageJ software [42]. Regarding the floral stems, the length, the number of cauline leaves, the mass and the diameter at the base of the floral stem were measured before freezing. Data are described in [43].

### 2.3. Histological Staining of CWs

Whole rosettes and base of floral stems were harvested in 50 mL Falcon tubes, and rapidly infiltrated under vacuum with FAA (10% formalin (37% formaldehyde solution, Sigma-Aldrich, Saint-Quentin Fallavier, France); 50% ethyl alcohol; 5% acetic acid; 35% distilled water). They were fixed for 16 h at 4 °C. The dehydration and paraplast infiltration protocol was described in [44]. The whole rosettes were processed and stained as described in [16] to keep the phyllotaxy intact. The stained sections were analysed using a Nanozoomer slide scanner (Hamamatsu, Shizuoka, Japan) to produce whole slide scan at 20× resolution. The scans were analyzed using NDP view (Hamamatsu) and the images were directly extracted from the viewer to assemble the figure.

### 2.4. Extraction of Proteins from Purified CWs

CW purification was performed as described [45]. We have carried out the sequential extraction of proteins from purified CWs as described, successfully using solutions containing CaCl_2_ 0.2 M and LiCl 2 M and buffered at pH 4.6 [46]. Typically, 0.2 g of lyophilized CWs was used for one extraction and about 500 µg proteins were extracted. The final protein extract was lyophilized. Proteins were quantified with the CooAssay Protein Assay kit (Interchim, Montluçon, France).

### 2.5. Sequential CW Polysaccharides Extraction and Identification

The sequential extraction of CW polysaccharides was performed in four steps and detailed in [16]. In summary, 100 mg of a deproteinized CW fraction were used. Four successive extractions were carried out to obtain extracts enriched in pectins (E1 and E2, respectively using CDTA 50 mM and 50 mM Na_2_CO_3_) and hemicelluloses (E3 and E4, respectively using 20 mM NaBH_4_ and 4 M NaOH). Each extract was hydrolyzed in 2 N TFA for 1 h at 120 °C. After 10× dilution in UHQ water, monosaccharides were analysed by High-Performance Anion-Exchange Chromatography coupled to Pulsed Amperometric Detection (HPAEC-PAD; Dionex, Sunnyvale, California, USA) using a CarboPac PA1 column (Dionex). L-Fuc, L-Rha, L-Ara, D-Gal and GalA (Sigma-Aldrich); D-Glc (Merck, Darmstadt, Germany); D-Xyl (Roche, Mannheim, Germany) were used as standard monosaccharides for identification and quantification. Data are described in [43]. CW polysaccharides reconstruction was performed using formula previously described [16,43] and adapted from [47].

### 2.6. Identification of Proteins by LC-MS/MS

The identification of proteins extracted from CWs was performed by LC-MS/MS at the PAPPSO proteomic platform (pappso.inrae.fr) after tryptic digestion in solution as described [48]. Parameters for MS data processing in the X!Tandem software (JACKHAMMER, 2013.6.15, www.thegpm.org/tandem/) and the X!Tandem Pipeline 3.3.4 [49] are detailed in [48]. Tryptic digestion was declared with no possible miscleavage. Only proteins identified with at least two different specific peptides in the same sample and found in at least two biological replicates were validated. Furthermore, quantification was performed on peptides with standard deviation retention times lower than 20 s and peak width lower or equal to 100 s. Raw data are described in [50].

### 2.7. RNA Sequencing

Protocols of the transcriptomic analysis were as described in [16]. RNA-Seq data and procedure details are available in Appendix A. Resulting sequences are available at NCBI short read archive (SRA, BioProject PRJNA344545). Raw data are described in [50].

### 2.8. Bioinformatics Annotation of Proteins and Quantification

The prediction of sub-cellular localization and functional domains of proteins was performed with the *ProtAnnDB* tool (www.polebio.lrsv.ups-tlse.fr/ProtAnnDB/) [51]. A protein was considered as a CWP if two bioinformatics programs predicted it as secreted, no intracellular retention signal was found and no more than one trans-membrane domain was found as described [52]. We have checked the prediction of the cell wall localization of many of the identified CWP families by doing a literature search as well as a mining of the *WallProtDB* database dedicated to cell wall proteomes (www.polebio.lrsv.ups-tlse.fr/WallProtDB). Quantification was only operated for CWPs using the MassChroQ software [53] as in [54]. The functional annotation of CWPs is given according to *WallProtDB*. All the LC-MS/MS data have been deposited at PROTICdb (http://proteus.moulon.inra.fr/w2dpage/proticdb/angular/#/projects/197) and CWP MS data are also available at *WallProtDB*.

### 2.9. Statistical Analysis and Data Integration

Data analysis and methodology were achieved as described [40]. We have performed multi-block supervised analyses with the sparse version Multi-Block Partial Least Squares Discriminant Analysis (MB-PLS-DA) implemented in the mixOmics R package version 6.1.1. (http://www.bioconductor.org/packages/release/bioc/html/mixOmics.html) [39]. They have taken into account all phenomics and metabolomics variables associated to a restrictive pool of 40 CWPs and 40 genes having modified transcript levels and encoding CWPs identified in the proteomic studies. To improve the visualization, the clustered image map was done with the high scoring 20 proteins and 20 genes and all the variables of the phenomics and metabolomics blocks.

## 3. Results

We have selected four *A. thaliana* populations (Roch, Grip, Hern and Hosp) among the 30 new populations collected in the Pyrenees Mountains recently characterized [38]. They constitute four independent homogeneous populations originating from various altitudes (Table 1) and they do not belong to the genetic cluster of Col (named III) which was taken as a reference ecotype. Roch and Grip belong to the same genetic cluster (named I), whereas Hern and Hosp belong to a third one (named II) (see [37] for the genetics characterization of the populations). In addition, within each genetic cluster, one population originates from moderate altitude (Roch and Hern) and the other one from high altitude (Grip and Hosp). The altitude clustering is also illustrated by the climate PC1 index that summarizes all the environmental and climatic data associated with each population.

### 3.1. The Morphological Phenotypes Are Temperature- and Organ-Dependent

The macro- and micro-phenotypes of rosettes of plants grown at 22 and 15 °C were analyzed. We have collected floral stems at equivalent degree days in order to be at the same developmental stage despite differential stem grow speed. The impact of the 15 °C growth temperature on rosettes was striking. The number of rosette leaves increased for all the Pyrenean population as well as for Col (e.g., a 60% increase for Col, Figure 1B), compared to that of plants grown at 22 °C.

Besides, the number of rosette leaves at the time of bolting could be contrasted: Col plants grown at 15 °C had the same number of rosette leaves as Roch grown at 22 °C and Hosp grown at 15 °C. The higher the altitude at which the Pyrenean populations were collected, the higher the number of rosette leaves prior bolting whatever the growth temperature (compare data in Figure 1B and Table 1). Except for Hern, an increase of the number of cauline leaves at the first flower-stage was observed for the 15 °C growth temperature (Figure 1C). Similarly, Hern had a tendency to show opposite floral stem traits (mass, diameter and length) as compared to the other populations grown at 15 °C (Appendix A).

The leaf number, combined with the mass, the diameter, the density and the projected area of rosettes (Appendix A) were integrated in the statistical study described thereafter (see § 2.5 and 2.6).

We have completed this macro-phenotyping with a microscopic phenotyping of the petioles of rosettes leaves. The petioles were cut into cross-sections (Figure 2). Two examples of the whole phyllotaxy present on the tissue arrays are given for Col (Figure 2U) and Roch (Figure 2V) grown at 15 °C to show the homogeneity of the staining pattern in a given condition. In a CW context, the astra blue/safranin red staining has a relative specificity for hydrophilic polysaccharides stained in blue and hydrophobic CW compounds stained in red.

At both 22 and 15 °C growth temperatures, the petiole of Col displayed a typical staining pattern with astra blue-positive cortical parenchyma CWs vs. safranin red-positive epidermis cuticle and vasculature lignified walls. At 22 °C, and in reference to Col cross sections, the safranin red staining was more intense in the epidermis cuticle of Grip, Hern and Hosp, and somehow in the vasculature of Grip and Hern, probably reflecting the corresponding cuticle reinforcement and vasculature increased lignification, respectively. At 15 °C, the safranin red staining was more intense in all the tissues of Roch, Grip, Hern and Hosp as compared to Col. These included the epidermis, the lignified vasculature as well as the cortical parenchyma. Similarly to Grip, Hern and Hosp grown at 22 °C, the epidermis and vasculature safranin red intense staining probably corresponded to cuticle reinforcement and vasculature increased lignification, respectively. However, the nature of the hydrophobic CW compounds responsible for the cortical parenchyma safranin reactivity was unclear.

### 3.2. The CW Composition of Aerial Organs Varies Depending upon Temperature Growth Conditions

We have analyzed the monosaccharide composition in rosettes and stems grown at 15 and 22 °C. We could reconstruct major CW polysaccharides as described [16,51]. Rosettes had higher ratios of linear to branched pectins than floral stems. Rosettes and floral stems contained lower ratios at 15 °C than at 22 °C (Figure 3). Furthermore, the ratio variations between the two growth temperatures were higher in floral stems than in rosettes for all the populations with the exception of Hern. We have performed a similar analysis for the other polysaccharides (Appendix A). To summarize, rosettes of plants grown at 15 °C contained more xyloglucans and a higher ratio of branched rhamnogalacturonans I than those of plants grown at 22 °C. Conversely, floral stems of Col, Grip and Hosp contained more xyloglucans when grown at 22 °C than at 15 °C. Interestingly, for rhamnogalacturonans I, homogalacturonans and to a lesser extend xyloglucans, differences between populations were more pronounced in floral stems at 15 °C than at 22 °C. Finally, homogalacturonans (also called linear pectins) were less abundant in rosettes and stems of plants grown at 15 °C than in those of plants grown at 22 °C.

### 3.3. The CW Proteome Varies at Sub-Optimal Growth Temperature

The CW proteomic analysis allowed identifying 364 and 414 different CWPs in rosettes and floral stems, respectively (Appendix A). Eighteen and 29 new CWPs were identified in this study as compared to the previously published *A. thaliana* Col and Sha rosette and floral stems proteomes, respectively [48,54]. We have found CWPs specifically accumulating in certain populations and/or at either growth temperature. For instance, AT5G59320 (AtLTP3) and AT5G59310 (AtLTP4) were more abundant at 15 °C, whereas AT4G37800 (AtXTH7) was more abundant at 22 °C in all the Pyrenean populations and Col. AT1G78450 (unknown function) was specific to Roch (see Appendix A for the detailed results).

Even if the principal component analysis (PCA) is an unsupervised method, it clearly highlighted groups of samples and allowed evaluating the good reproducibility of a given experiment [40], when performed on quantified CWPs (Figure 4). The analysis of the two organs showed a growth temperature-specific response of each population at the CW proteome level. Regarding rosettes, the first component (PC1) separated the samples according to the growth temperature (22 vs. 15 °C) and the second one (PC2) according to the population (Hosp vs. the other populations) (Figure 4A). For floral stems, PCA clustered together the samples according to the genetic clusters defined above: clusters III (Col) and I (Roch/Grip) vs. cluster II (Hern/Hosp) (Figure 4B). The growth temperature affected less clusters III and I than cluster II as shown by the better clustering of Col, Roch and Grip compared to Hern and Hosp. Col, Roch, Grip and Hosp grown at 22 °C were located farther right on the PC1 axis as compared to the corresponding populations grown at 15 °C. Conversely, the Hern samples showed a reverse location with regard to PC1 compared to the other populations.

### 3.4. Regulation of Gene Expression Depends on Both Growth Temperatures and Populations

RNA-Seq analyses allowed the identification of 19,763 and 22,570 genes with significant levels of expression in rosettes and floral stems respectively (see raw data and statistical analyses in Appendix A). As for CW proteomics data, the unsupervised multivariate analysis of the “whole transcriptome” of rosettes (Figure 5A) and floral stems (Figure 5B) showed the reproducibility of the data by the clustering of the biological replicates.

The transcript levels of the genes encoding CWPs identified by the CW proteomic analysis were extracted (Appendix A) in order to correlate proteomics and transcriptomics data. PCA restricted to these transcripts (defined as “CW transcriptome”) showed similar separation as that obtained with the “whole transcriptomes” (compare Figure 5A with Figure 5C; Figure 5B with Figure 5D), as observed in a previous study [16].

For rosettes, the analysis brought out the strong effect of the temperature that separated the samples on PC1 (Figure 5A,B). The PCAs of “whole transcriptomes” and “CW transcriptomes” of rosettes both isolated Hosp grown at 15 °C. A closer analysis indicated that the set of genes explaining the discrimination of Hosp grown at 15 °C was not related to the CW proteomes (Appendix A). In turn, the analysis focused on the CWP transcriptomes highlighted a few candidates such as *AT2G25510* (unknown function), *AT2G14610* (Cys-rich secretory protein) which were overexpressed or *AT5G26000* (glycoside hydrolase family 1) which was down-regulated. In floral stems, PCA highlighted the particular behaviors of Hern and Col which were both separated from the other samples (Figure 4B and Figure 5B,D; Appendix A).

The impact of the growth temperature on gene expression levels was observed in both rosettes and floral stems for the Pyrenean populations and for Col and allowed discriminating genes exhibiting specific expression patterns.

### 3.5. An Integrative Study Highlights the Different Responses of the Rosettes and Floral Stems According to the Growth Temperature

Thereafter, we refer to each omics dataset (phenomics, metabolomics, proteomics and transcriptomics) as a block and we have used a previously described framework for the integration of the omics data [40].

To provide an insight into the results of the integrative process, we have considered the pairwise correlations between each pair of blocks [39]. The higher the values, the stronger the relationships between the blocks (Figure 6A). In rosettes, these values were higher than 0.83 (value obtained between CW transcriptomics and phenomics). These values were mainly due to a strong temperature effect as indicated in the segregation of two groups in Appendix A. The highest correlation was found between the CW transcriptomics and CW proteomics blocks (0.97). All the blocks showed a clear segregation between the growth temperatures according to the first component that confirmed the strong effect of the temperature (Figure 6A and Supplementary Figure 5A).

Using the sparse multi-block partial least squares discriminant analysis (MB-PLS-DA) analysis, we could identify the variables from each block involved in the discrimination according to the growth temperature (Appendix A and Appendix A). The clustered image map also includes the results of a hierarchical clustering performed jointly on both the variables and the samples. Again, the samples were clearly discriminated according to growth temperatures. The list of discriminated transcripts and CWPs is given in Table 2. Among this set of CW transcripts and CWPs, 23% corresponded to proteins acting on CW polysaccharides, 18% to proteases and 18% to proteins possibly related to lipid metabolism. Besides, most changes occurred at the 15 °C growth temperature (30 out of the 40 candidates).

For floral stems, we have observed lower correlations between block pairs (Appendix A). As for rosettes, we have found the highest correlation (0.9) between the proteomics and CW transcriptomics blocks and the lowest (0.63) between the phenomics and metabolomics blocks. Low correlation could be due to the weak segregation between the growth temperatures in the first component obtained in the phenomics and metabolomics blocks. Table 3 provides an overview of the CW transcripts and CWPs whose levels of accumulation were modified upon different growth temperatures. Among these CW transcripts and CWPs, 22% corresponded to oxido-reductases and 22% to proteins possibly related to lipid metabolism.

The correlation observed between block pairs was probably lowered by the specific phenotype of Hern floral stems at 15 °C (Appendix A). The important effects of the growth temperature and the specificity of Hern were also highlighted by the hierarchical clustering of the samples in the clustered image map of the discriminating variables (Appendix A). The dendrogram discriminated the samples on the basis on growth temperatures: Roch, Grip, Col and Hosp grown at 15 °C clustered together whereas all the plants grown at 22 °C together with Hern grown at 15 °C formed another cluster.

### 3.6. The Hosp and the Hern Populations Exhibit Particular Phenotypic Responses Regarding Rosette and Floral Stem Development

The integrative analysis has allowed the identification of relevant candidate genes/proteins related to the temperature effect. The next step was to analysis the specific responses of the different populations at the level of the two analyzed organs, rosettes and floral stems.

For rosettes, when we have compared the metabolomics and the phenomics blocks, the correlation varied from 0.87 between the CW proteomics and CW transcriptomics blocks to 0.56 were compared (Figure 7A). Regarding the individual plots, each block discriminated different populations (Supplementary Figure 5B): (i) in the CW proteomics block, we have observed three clusters, namely Roch/Grip, Col/Hern and Hosp; (ii) we have found a similar clustering in the CW transcriptomics block with a better segregation; (iii) the phenomics block only allowed a separation of Roch at 22 °C; (iv) the metabolomics block did not allow distinguishing the populations and Col. These differences between the Pyrenean populations profiles resulted from the contrasted values of the variables in each block represented in Appendix A. 

The clustered image map of the samples for each block involved in the discrimination according to the Pyrenean populations and Col showed a segregation into three groups (Figure 7B). From top to bottom, the first one contained only Hosp, the second one Roch grown at both temperatures and Grip grown at 15 °C, and the third one was composed of Hern and Col grown at both temperatures and Grip grown at 22 °C. The specific profile of Hosp was supported by a pool of CW transcripts and CWPs sorted in Table 4 (see also Appendix A). One fourth of them corresponded to proteins possibly involved in lipid metabolism and 27% to proteins of yet unknown function.

For floral stems, the correlations between components from each block were highly variable from 0.95 between the CW proteomics and CW transcriptomics blocks to 0.25 between the metabolomics and the phenomics blocks (Appendix A). This large range of correlation was due to the good clustering of the samples observed between the CW proteomics and CW transcriptomics blocks and the poor clustering and large variability observed for the metabolomics and the phenomics blocks (Appendix A). This observation was confirmed by the strong segregation of Hern along the first axis and of Hosp along the second one regarding the CW proteomics and CW transcriptomics blocks on the individual plots (Appendix A). On the other hand, only Hern could be separated between the two growth temperature conditions with the phenomics block, but not as clearly as with the CW proteomics and CW transcriptomics blocks. At the end, the Pyrenean populations and Col were well separated by a few discriminating variables (Appendix A).

Altogether, this analysis allowed segregating the samples in three major groups: (i) Hern, (ii) Hosp, and (iii) Col, Grip and Roch. Using the sparse approach, we could sort a restricted pool of transcripts and CWPs differentially accumulated in floral stems of Hosp and Hern vs. Col, Grip and Roch (Table 5, Appendix A). For Hosp, the CW transcripts and proteins mainly corresponded to proteins acting on cell wall polysaccharides (35%), whereas for Hern, proteases and proteins acting on cell wall polysaccharides represented 42% and 35% of the set, respectively.

## 4. Discussion

A comparative study of four populations of *A. thaliana* originated from a Pyrenees Mountains altitudinal gradient and of the well-described Col ecotype was performed to better understand the CW plasticity in response to environmental changes. We have analyzed the plant response to sub-optimal growth conditions, i.e., at 15 °C, with an integrative approach using heterogeneous omics datasets to evaluate the impact on the CW. Altogether, we could show (i) a major effect of the growth temperature, (ii) the organ specificity of the response, and (iii) the particularities of Hosp and Hern, compared to Grip, Roch and Col. Since many CW transcripts and CWPs were highlighted in this work, we have focused on a few of them in the following discussion.

### 4.1. Temperature Has an Important Impact on Plant Development and on the CW

The observed macro-phenotypes have highlighted the ubiquitous impact of the growth temperature on the development of both rosettes and floral stems for all plants. The petioles of the plants grown at 15 °C, especially the Pyrenees Mountains population, presented a more intense staining of hydrophobic compounds in the cuticle epidermis of the cross-sections and around vessels. In the former case, such compounds could be present in the cuticle. This would be consistent with the observations reported in a previous study performed on the Sha ecotype originated from 3400 m in a high valley of Tajikistan [16]. Indeed, when grown at 15 °C, Sha petioles showed a thicker cuticle than Col petioles. In the latter case, these hydrophobic compounds could be lignin and ferulic acid which could contribute to an increase of CW rigidity [18].

Although some trends are common to rosettes and floral stems upon growth at 22 °C or 15 °C, only two CW transcripts/CWPs of the oxido-reductases functional class were common to the short list of those differentially accumulated at a given growth temperature: AT5G15350 which is an early nodulin (AtEN22) homologous to blue copper binding proteins, and AT1G41830 which is a multicopper oxidase (AtSKS6) homologous to the *A. thaliana* SKU5 (SKEWED ROOT 5). To our knowledge, the role of the former protein has not been studied yet whereas SKU5 was shown to play a role in root directional growth [55] and more recently in the spaceflight response [56].

To summarize, the impact of the growth temperature on the levels of accumulation of CW polysaccharides, CW transcripts or CWPs was observed in both rosettes and floral stems for all plants. However, this response was mostly organ-specific.

### 4.2. The Response to Growth Temperature is Organ-Specific

The CW structure and the composition of rosettes and floral stems were very different in the two growth conditions. For example, at the 15 °C growth temperature, we have found more xyloglucans and more branched rhamnogalacturonans I in the rosettes of all plants whereas we have observed less xyloglucans in the floral stems of Col, Grip and Hos. Modifications in the composition of polysaccharides were already observed in *Miscanthus giganteus* ecotypes upon cold stress. In particular, the amount of (1,3)(1,4)-β-d-glucans (mixed-linked glucans) was increased [57]. In tobacco pollen tubes, cold was shown to induce changes in the pattern of deposition of homogalacturonans (both methylated and demethylated), callose and cellulose [58]. Such changes are assumed to modify the CW properties. CWPs possibly involved in these modifications could not be identified among the identified candidates maybe because the observed changes have occurred at earlier stages of development. Indeed, the plants have been continuously grown either at 22 °C or at 15 °C. We do not apply a stress as it is usually defined, i.e., a sudden change in the environmental conditions [16]. The observed changes could also occur at the level of the biosynthesis of the CW polysaccharides, which takes place in the Golgi apparatus and which could not be seen in this study.

Considering the temperature as the factor for discriminant analysis, the integrative analysis of rosettes highlighted an enrichment in three CWPs functional classes (proteins acting on CW polysaccharides, proteases and proteins related to lipid metabolism) among the candidate CW transcripts and CWPs (Appendix A, Table 2). Most of the proteins acting on CW polysaccharides were glycoside hydrolases (GHs). Three GH27 transcripts encoding α-galactosidases (*AT3G26380*, *APSE*; *AT5G08370*, *AGAL1*; *AT5G08380*, *AGAL2*) were up-regulated at 15 °C. An ortholog of *AT5G08380* in *Thlaspi arvense*, another *Brassicaceae*, was shown to be up-regulated during cold acclimation [59,60]. Besides, APSE, and to a lesser extent AGAL1 and 2, have been assumed to be involved in the hydrolysis of the β-l-Arap residues of the glycan moiety of arabinogalactan proteins in muro [61]. Among proteases, one Ser carboxypeptidase (AtSCPL10) was accumulated at the 15 °C growth temperature and transcripts encoding AtSCPL29 and 49 were more abundant. Recently, a protein of the same family (AtSCPL41) was shown to play a role in membrane lipid metabolism [62]. Besides ASPG1 (ASPARTIC PROTEASE IN GUARD CELL 1) was shown to be involved in drought avoidance via ABA-dependent signaling [63]. Several proteins possibly involved in lipid metabolism, three non-specific lipid transfer proteins (LTPs) were found to be more abundant at 15 °C than at 22 °C. *AtLTP3* (*AT5G59320*) was previously shown to be involved in the response to freezing stress [64] and was part of a tandem duplication with *AtLTP4* (*AT5G59310*). The two corresponding proteins have been found in close proximity in the dendrogram (correlation > 0.9, Appendix A). Even if no study has shown the role of these LTPs in muro, their accumulation in rosettes at 15 °C could be related to the high number of hydrophobic compounds found in the cuticle epidermis of the petiole cross-sections mentioned above. However, the *atltp3* mutant did not show any cuticular phenotype in normal growth conditions, but higher sensitivity to drought stress, whereas overexpressing plants were more tolerant to freezing and drought [64]. The levels of expression of *AtLTP4* and *AtLTP8* were increased in the *atltp3* mutant, suggesting functional redundancy [64]. The analysis of double or triple mutants could contribute to a better understanding of the impact of low temperature growth on cuticle formation.

In floral stems, the majority of the candidate CW transcripts and CWPs identified at the 15 °C growth temperature belonged to the oxido-reductases and proteins possibly involved in lipid metabolism functional classes. Two oxido-reductases families are well-represented: blue copper binding proteins (AT5G15350, AtEN22; AT4G31840, AtEN13; AT4G12880, AtEN20) and multicopper oxidases (AT5G21100; AT3G13990, AtSKS11; AT1G41830, AtSKS6). As mentioned above, AtEN22 and AtSKS6 are the two candidates common to rosettes and floral stems. Regarding proteins possibly involved in lipid metabolism, *AtLTPg4* (*AT1G27950*) and *AT5G62210* encoding a protein of yet unknown function, were highlighted. Interestingly, the *AT5G62210* transcript and the encoded protein exhibiting a PLAT/LH2 domain both accumulated at 15 °C. This protein showed 41% homology with that encoded by the pathogen-induced *CAPIP2* gene which was described as a response protein against abiotic stresses [65,66]. A previous transcriptomics analysis of epidermal peels of *A. thaliana* stems has previously highlighted genes encoding *LTPs* (*AtLTP2*, *5* and *7* and *AtLTPg4*, *6*, *7* and *21*) and thus allowed identifying new proteins possibly playing roles in the biosynthesis of waxes and cutin [67]. The importance of the cuticle of the floral stems is critical as it constitutes a vital hydrophobic barrier, providing mechanical strength and viscoelastic properties as well as protecting the plant from the environmental stresses [25].

Altogether, the growth temperature had a different effect on rosettes and floral stems at the molecular level. Distinct CW transcripts and CWPs of interest could be highlighted. However, in both cases, the proteins possibly related to lipid metabolism were particularly well-represented, suggesting a critical role of the cuticle in the acclimation to low growth temperatures.

### 4.3. The Hern and Hosp Populations Exhibit Particularities in Their Responses to Growth Temperature

Hosp showed different responses at the level of rosettes at the 15 °C growth temperature. Six CW transcripts and CWPs allowing the discrimination of Hosp had unknown function, like *AT5G38980* for which both transcripts and proteins were accumulated. AtLTPg6 (AT1G55260) was more abundant and *AtLTP6* (*AT3G08770*) transcripts were up-regulated. Interestingly, the transcripts levels of these two candidates were also up-regulated in the Sha altitudinal ecotype grown at 15 °C [16] and LTPgs have been reported to be implicated in the transport of lipids through the hydrophilic CWs to build up the cuticle [68]. These results are consistent with the higher level of hydrophobic compounds observed in the epidermis cuticle of Hosp petioles compared to Col.

The behavior of Hern and Hosp was different from that of Grip, Roch and Col with regard to floral stems. However, we could not identify any common candidates when the integrative analysis was focused on the temperature parameter. The specificity of Hern floral stems could be noticed by the clustering of all the phenomics and metabolomics variables, except the ratio of rhamnogalacturonan I branching (Appendix A). Co-expressed with these variables, six proteins were more abundant in Hern, three of them belonged to the proteins with interaction domains functional class and two to the proteins acting on CW polysaccharides functional class. Among the proteins with interaction domains, the two lectin genes (*AT3G15356*, *AT3G16530*) were known to be sensitive to abiotic stress and up-regulated upon hormonal treatments [69,70]. *AT3G04720* (*PR4*) was positively regulated by jasmonate and ethylene [71]. *AT4G16260* encoding a β-1,3-glucosidase belonging to the proteins acting on CW polysaccharides functional class was also regulated by ethylene [72]. These results highlight a possible link between the Hern floral stem phenotype and the jasmonate and ethylene hormones. Hormones were not analyzed in this study but could play a major role in the phenotypic plasticity of floral stems. Interestingly, transcripts encoding proteins acting on CW polysaccharides and proteases which were well-represented in this set of candidates were down-regulated in Hern. Proteases play roles in the generation of signals involved in plant development, in the maturation of proteins and in protein turnover [73,74]. These processes which are still poorly understood could be important in the phenotypic differentiation of Hern floral stems at 15 °C. Hern appears as an interesting population to study the contribution of such proteins to CW plasticity in floral stems. Besides, Hern floral stems exhibited lower levels of six transcripts among which two encoded proteases: AT1G28110 (AtSCPL25) and AT4G00230 (AtSBT4.14).

Hosp also showed particularities at the proteomics and transcriptomics level. With less contrasted altered phenotypic traits than Hern, Hosp exhibited a lower number of CW transcripts and CWPs with modified levels of accumulation. Nevertheless, some candidates CW transcripts or CWPs appeared to specifically accumulate in Hosp such as the Ser (AT2G12480, AtSCPL43) and the Asp (AT3G02740) proteases. In addition, the *AT1G78820* transcripts encoding a lectin were accumulated twice as much as in Col. Moreover, the transcripts of two candidates already identified in the stem epidermis [66] were specific to Hosp and were found to be more abundant at 15 °C: *AtLTPg4* (*AT1G27950*) and *PME41* (*PECTIN METHYLESTERASE 41*, *AT4G02330*). Mutants impaired in *AtLTPg4* and *AtLTPg21* exhibit significant alterations in the cuticular wax composition [75]. Also, the *pme41* mutants were shown to be sensitive to chilling stress and this phenotype could be related to the brassinosteroids responses [75]. This PME could play a role in plant chilling tolerance by modifying the mechanical properties of the CW and specifically the esterification rate of homogalacturonans [76]. Beyond, the possible regulation of the PMEs by hormones such as brassinosteroids could provide further insight into their role in the acclimation of plants to low temperature.

## 5. Conclusions

The aim of this study was to identify new players of the CW plasticity upon acclimation to sub-optimal growth temperatures by an integrative study using different *A. thaliana* populations collected in the Pyrenees Mountains. A contrasted CW plasticity has been observed in rosettes and in stems such as the increase of xyloglucan content in the former and their decrease in the latter at the 15 °C growth temperature. The differential CW plasticity responses between organs could also be illustrated by the differences observed between the CW transcriptomes and proteomes depending on both the population and the growth temperature. The global integration of all four blocks of data (phenomics, metabolomics, proteomics and transcriptomics) supervised by a categorical outcome (temperature or ecotype) has allowed finding new candidates for functional studies, such as LTPs which could play roles in the biogenesis of the cuticle. Such candidates would not have been identified using a mono-block analysis.

This work has revealed that low temperature acclimation affects the expression of many genes at the transcriptional or post-transcriptional levels, many of them having yet unknown biological functions. Importantly, this powerful integrative analysis has allowed identifying interesting candidates for exploring the adaptation of natural populations to low temperature acclimation. Further studies using mutants with altered expression of these candidates are now necessary to elucidate their biological significance. In this way, hypothesis-based approaches exploring understudied traits can provide new issues in *A. thaliana* research. Furthermore, this knowledge could be transferred to plants of agronomical interest.

## Figures and Tables

**Figure 1 cells-09-02249-f001:**
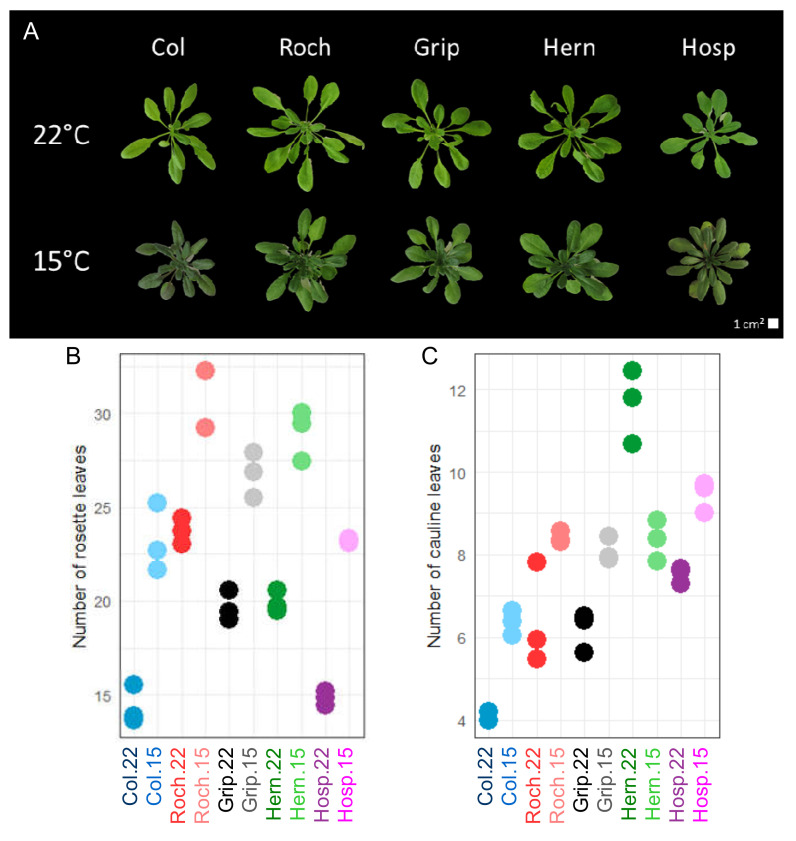
Macro-phenotyping of Col, Roch, Grip, Hern and Hosp grown at 22 or 15 °C. (**A**) Pictures of rosettes at the time of sampling; (**B**) number of rosette leaves at the bolting stage; (**C**) number of cauline leaves at the first flower-stage. Twenty plants from 3 independent batches were analyzed and each dot stands for the mean value for one batch.

**Figure 2 cells-09-02249-f002:**
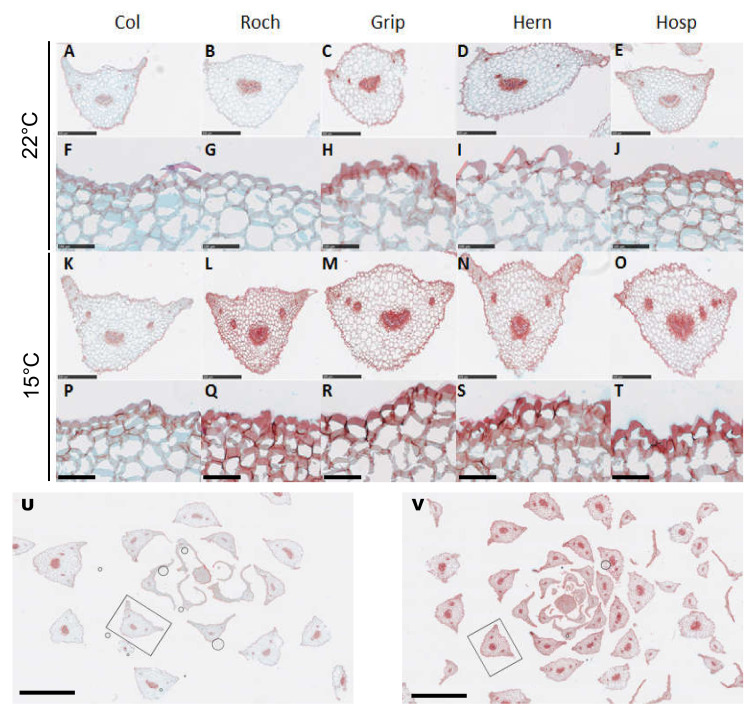
Phenotyping of leaf petioles of Col, Roch, Grip, Hern and Hosp at the bolting stage after 4 or 6 weeks of culture at 22 °C (**A**–**J**) or 15 °C (**K**–**T**). Cross-sections were stained with safranin red and astra blue. Representative plant petioles among more than 30 petioles from independent batches are shown. The black frames in (**U**,**V**) show the petiole cross-section displayed in (**K**,**L**) at a higher magnification, respectively. Scale bars represent 500 µm (**A**–**E**, **K**–**O**), 100 µm (**F**–**J**, **P**–**T**) or 2 mm (**U**, **V**).

**Figure 3 cells-09-02249-f003:**
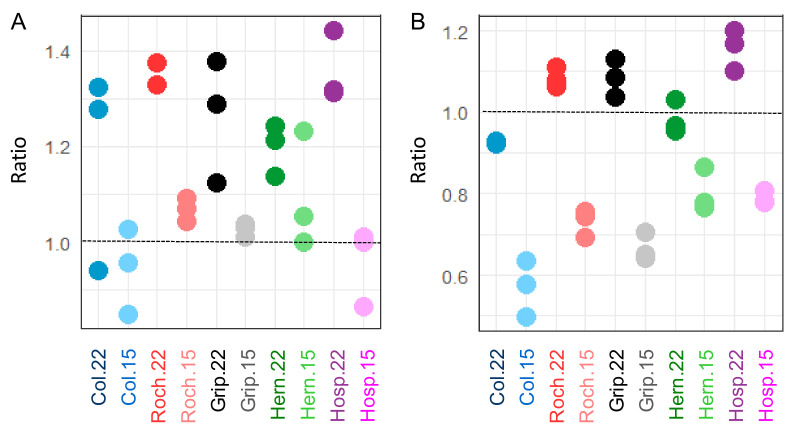
Ratios of linear to branched pectins of Col, Roch, Grip, Hern and Hosp grown at 22 or 15 °C: (**A**) rosettes; (**B**) floral stems. Mean values calculated from 3 independent batches are represented.

**Figure 4 cells-09-02249-f004:**
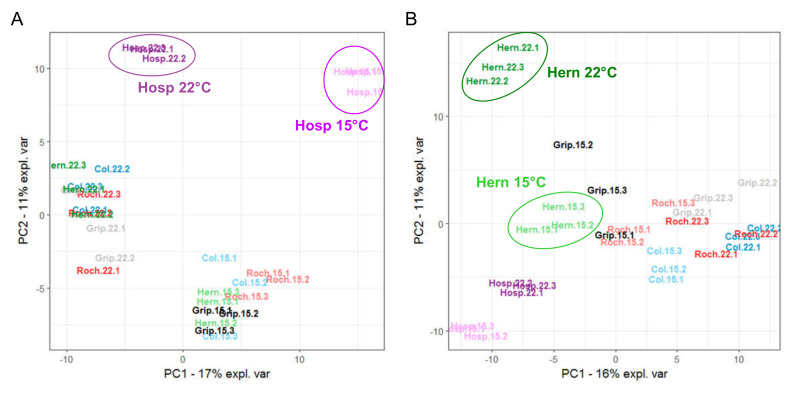
Overall comparison of the cell wall (CW) proteomes of Col, Roch, Grip, Hern and Hosp grown at 22 °C (bright colors) or 15 °C (pale colors). Scaled PCA of the proteomic quantitative analysis: (**A**) rosettes; (**B**) floral stems. The ellipses focus on the specific profiles discussed in the Results section.

**Figure 5 cells-09-02249-f005:**
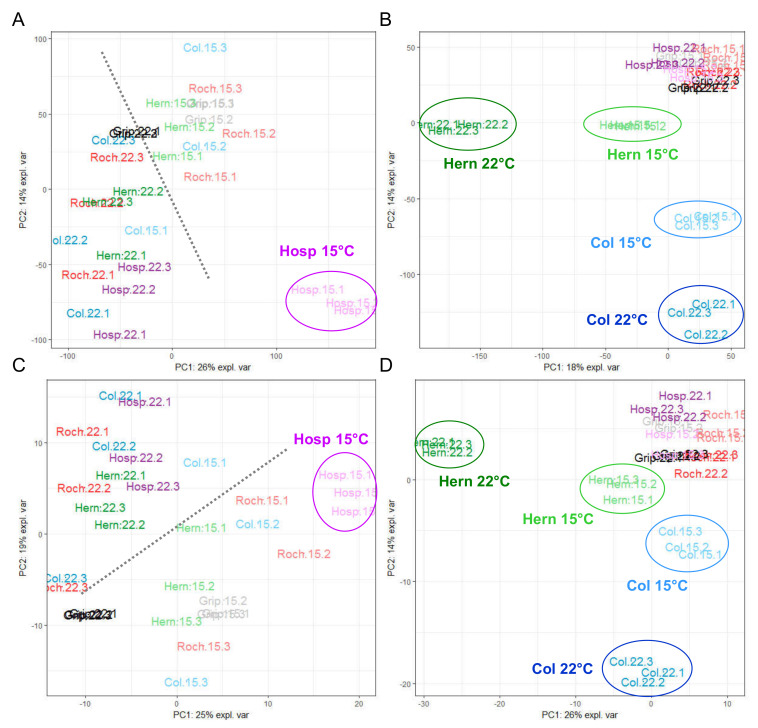
Overall comparisons of the transcriptomes of Col, Roch, Grip, Hern and Hosp grown at 22 °C (bright colors) or 15 °C (pale colors). Scaled PCA of the “whole transcriptome”: (**A**) rosettes; (**B**) floral stems. Scale PCA of the “CW transcriptome”: (**C**) rosettes; (**D**) floral stems. The ellipses focus on the specific profiles discussed in the Results section. The grey dotted lines indicated the separation of the samples according to the growth temperature.

**Figure 6 cells-09-02249-f006:**
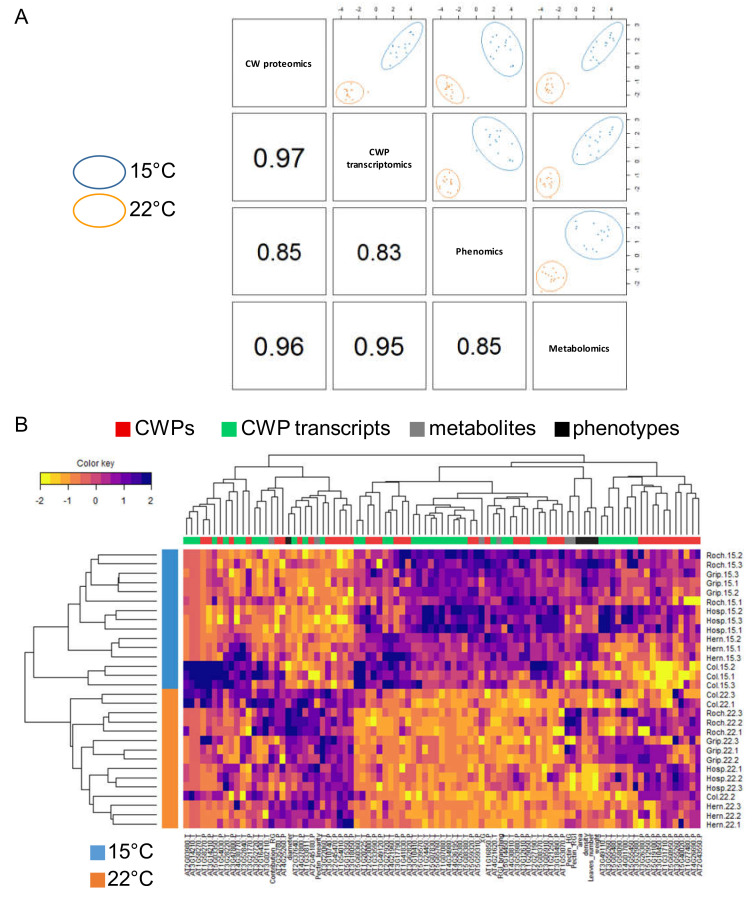
Graphical representation of the sparse multi-block partial least squares discriminant analysis (MB-PLS-DA) analysis discriminating the rosette samples of Col, Roch, Grip, Hern and Hosp according to the growth temperature. (**A**) The plotDIABLO shows the correlation inside each block pair; (**B**) clustered image map representing the multi-omics profiles for each sample discriminated by the growth temperature. The levels of yellow (lower values) and purple (higher values) denote scaled values for each variable. Note that colors are scaled per line.

**Figure 7 cells-09-02249-f007:**
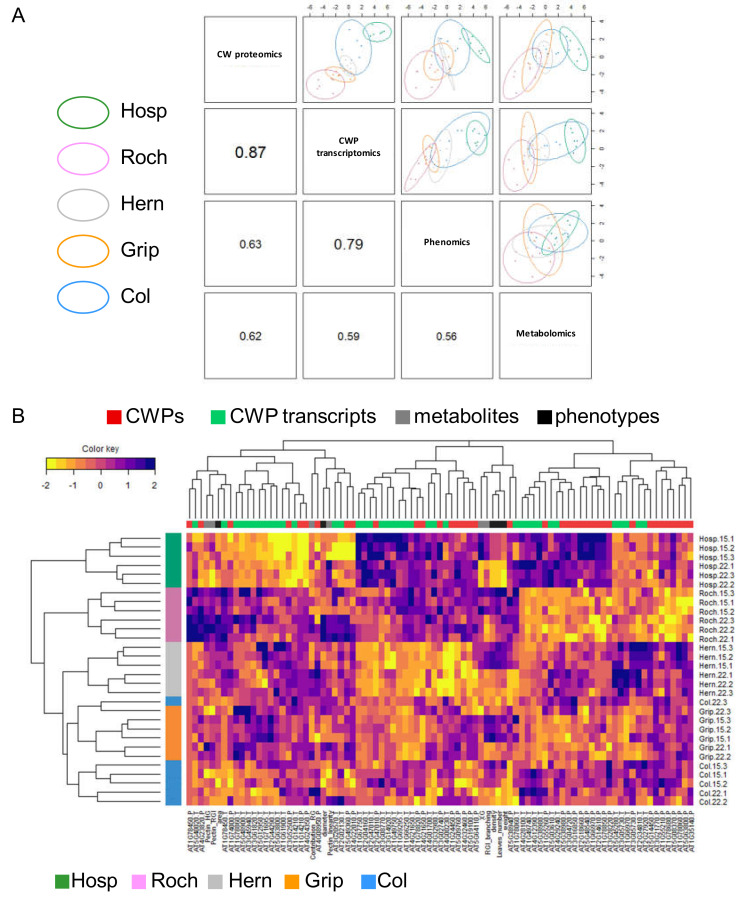
Graphical representation of sparse MB-PLS-DA analyses that discriminate the rosette samples according to Col, Roch, Grip, Hern and Hosp. (**A**) The plotDIABLO shows the correlation inside each block pairs; (**B**) clustered image map representing the multi-omics profiles for each sample discriminated according to populations and Col. The levels of yellow (lower values) and purple (higher values) denote scaled values for each variable. Note that colors are scaled per line.

**Table 1 cells-09-02249-t001:** Environmental and genetic parameters of *A. thaliana* populations.

Accession Name	Altitude (MASL) ^1^	Genetics Cluster ^2^	Climate PC1 ^3^
Col	200 ^4^	III	3
Roch	696	I	0.7
Grip	1190	I	−1.4
Hern	780	II	0.7
Hosp	1424	II	−1.6

^1^ MASL: Meters Above Sea Level. ^2^ The genetics clusters refer to the previous characterization of the Pyrenean populations [38]. ^3^ Climate PC1 (index that describes and explains the climate gradients).^4^ Altitude at which Col was originally collected.

**Table 2 cells-09-02249-t002:** Synopsis of CW proteins (CWPs) or CW transcripts differentially accumulated at a given growth temperature in rosettes.

AGI Code	Functional Class ^1^	Putative Function	Proteins ^2^	Transcripts ^2^	22 °C ^2^	15 °C ^2^
AT2G45180	LM	homologous to non-specific lipid transfer protein	+		+	
AT1G54010	LM	lipase acylhydrolase (GDSL family)	+		+	
AT5G15350	OR	early nodulin (AtEN22, AtENODL17), blue copper binding protein	+		+	
AT4G37800	PAC	GH16 (endoxyloglucan transferase) (AtXTH7)	+		+	
AT3G10740	PAC	GH51 (α-arabinofuranosidase)	+		+	
AT2G45470	S	fasciclin-like arabinogalactan protein (AtFLA8)	+		+	
AT3G18050	UF	expressed protein	+		+	
AT1G53070	ID	lectin (legume lectin domain)	+			+
AT1G33590	ID	expressed protein (LRR domains)	+			+
AT3G12610	ID	expressed protein (LRR domains)	+			+
AT3G62820	ID	plant invertase/pectin methylesterase inhibitor (PMEI)	+			+
AT5G59320	LM	non-specific lipid transfer protein (AtLTP1.2, AtLTP3)	+			+
AT5G59310	LM	non-specific lipid transfer protein (AtLTP1.11, AtLTP4)	+			+
AT3G53980	LM	non-specific lipid transfer protein (AtLTPd7)	+			+
AT1G41830	OR	multicopper oxidase (AtSKS6, homologous to SKU5)	+			+
AT2G23000	P	Ser carboxypeptidase (AtSCPL10)	+			+
AT3G18490	P	Asp protease (pepsin family) (ASPG1)	+			+
AT1G29050	PAC	homologous to *A. thaliana* PMR5 (carbohydrate acylation)	+			+
AT1G21250	S	receptor kinase (AtWAK1, WALL-ASSOCIATED KINASE 1)	+			+
AT1G16850	UF	expressed protein	+			+
AT1G33811	LM	lipase acylhydrolase (GDSL family)		+	+	
AT2G37640	PAC	α-expansin (ATHEXP ALPHA 1.9) (AtEXPA3)		+	+	
AT3G22060	UF	expressed protein		+	+	
AT5G14450	LM	lipase acylhydrolase (GDSL family)		+		+
AT2G27190	M	purple acid phosphatase (AtPAP12)		+		+
AT1G24450	M	ribonuclease III		+		+
AT1G07080	OR	thiol reductase (GILT family)		+		+
AT4G21960	OR	class III peroxidase (AtPrx42)		+		+
AT4G36195	P	Pro-Xaa carboxypeptidase (Peptidase family S28.A26, MEROPS)		+		+
AT5G65760	P	Pro-Xaa carboxypeptidase (Peptidase family S28.A02, MEROPS)		+		+
AT4G34980	P	Ser protease (AtSBT1.6)		+		+
AT3G10410	P	Ser carboxypeptidase (AtSCPL49)		+		+
AT4G30810	P	Ser carboxypeptidase (AtSCPL29)		+		+
AT3G26380	PAC	GH27 (α-galactosidase/melibiase, APSE)		+		+
AT5G08370	PAC	GH27 (α-galactosidase/melibiase, AGAL1)		+		+
AT5G08380	PAC	GH27 (α-galactosidase/melibiase, AGAL2)		+		+
AT5G07830	PAC	GH79 (endo-β-glucuronidase/heparanase)		+		+
AT3G05910	PAC	CE13 (pectin acylesterase - PAE) (AtPAE12)		+		+
AT4G21620	UF	expressed protein		+		+
AT5G39570	UF	expressed protein		+		+

The statistical analysis of the data was performed with a sparse MB-PLS-DA and selected the 20 best hit among each block. ^1^ ID: proteins with interaction domains (with proteins or polysaccharides); LM: proteins possibly related to lipid metabolism; M: miscellaneous proteins; OR: oxido-reductases; P: proteases; PAC: proteins acting on CW polysaccharides; S: proteins possibly involved in signaling; UF: proteins of yet unknown function. ^2^ “+“corresponds to proteins or transcripts showing an increased level of accumulation at a given growth temperature. The fold changes can be read in Appendix A.

**Table 3 cells-09-02249-t003:** Synopsis of CWPs or CW transcripts differentially accumulated at a given growth temperature in floral stems.

AGI Code ^1^	Functional Class ^2^	Putative Function	Proteins ^3^	Transcripts ^3^	22 °C ^3^	15 °C ^3^
AT2G10940	LM	homologous to non-specific lipid transfer protein	+		+	
AT5G21100	OR	multicopper oxidase	+		+	
AT5G15350	OR	early nodulin (AtEN22, AtENODL17), blue copper binding protein	+		+	
AT1G47128	P	Cys protease (papain family) (Peptidase family C01.064) (RD21A)	+		+	
AT4G21640	P	Ser protease (AtSBT3.15)	+		+	
AT3G10450	P	Ser carboxypeptidase (AtSCPL7)	+		+	
AT5G19740	P	peptidase M28 (peptidase family M28.A02, MEROPS)	+		+	
AT1G26560	PAC	GH1 (β-glucosidase) (AtBGLU40)	+		+	
AT2G06850	PAC	GH16 (endoxyloglucan transferase) (At-XTH4)	+		+	
AT1G10640	PAC	GH28 (polygalacturonase)	+		+	
AT3G02880	S	leucine-rich repeat receptor protein kinase (LRR III subfamily)	+		+	
AT3G22060	UF	expressed protein (Gnk2-homologous domain)	+		+	
AT5G62210	LM	expressed protein (PLAT/LH2 domain)	+			+
AT1G31550	LM	lipase acylhydrolase (GDSL family)	+			+
AT1G29670	LM	lipase acylhydrolase (GDSL family)	+			+
AT1G18250	M	thaumatin (PR5, ATLP1)	+			+
AT5G51950	OR	expressed protein (GMC oxido-reductase domain)	+			+
AT4G31840	OR	early nodulin (AtEN13, AtENODL15), blue copper binding protein	+			+
AT4G01700	PAC	GH19	+			+
AT4G23820	PAC	GH28 (polygalacturonase)	+			+
AT5G06860	ID	PGIP1 (LRR domains)		+	+	
AT3G13990	OR	multicopper oxidase (AtSKS11, homologous to SKU5)		+	+	
AT5G20230	OR	stellacyanin AtSTC1, BCB (blue copper binding protein)		+	+	
AT1G71695	OR	class III peroxidase (AtPrx12)		+	+	
AT2G39850	P	Ser protease (AtSBT4.1)		+	+	
AT4G21585	LM	phospholipase C/P1 nuclease		+		+
AT2G38540	LM	non-specific lipid-transfer protein (AtLTP1.5, AtLTP1)		+		+
AT1G27950	LM	non-specific lipid transfer protein (AtLTPg4, LTPG1)		+		+
**AT5G62210**	LM	expressed protein (lipase/lipooxygenase domain, PLAT/LH2)		+		+
AT2G04570	LM	lipase acylhydrolase (GDSL family)		+		+
AT1G41830	OR	multicopper oxidase (AtSKS6, homologous to SKU5)		+		+
AT4G12880	OR	early nodulin (AtEN20, AtENODL19), blue copper binding protein		+		+
AT2G33530	P	Ser carboxypeptidase (AtSCPL46)		+		+
AT5G45280	PAC	CE13 (pectin acylesterase - PAE) (AtPAE11)		+		+
AT4G38400	PAC	expansin-like A (ATHEXP BETA 2.2) (AtEXLA2)		+		+
AT5G55730	S	fasciclin-like arabinogalactan protein (AtFLA1)		+		+
AT2G04780	S	fasciclin-like arabinogalactan protein (AtFLA7)		+		+
AT3G06035	UF	expressed protein		+		+
AT4G00860	UF	expressed protein (DUF1138)		+		+
AT3G05730	UF	expressed protein		+		+

The statistical analysis of the data was performed with a sparse MB-PLS-DA and selected the 20 best hit among each block. ^1^ AGI codes in bold means that the protein and the corresponding transcripts were both differentially accumulated. ^2^ See legend to Table 2, for the meaning of the following abbreviations: ID, LM, M, OR, P, PAC, S, UF. ^3^ “+” corresponds to proteins or transcripts showing an increased level of accumulation at a given growth temperature. The fold changes can be read in Appendix A.

**Table 4 cells-09-02249-t004:** Synopsis of CWPs or CW transcripts differentially accumulated at a given growth temperature in floral stems.

AGI Code ^1^	Functional Class ^2^	Putative Function	Proteins ^3^	Transcripts ^3^
AT1G78850	ID	lectin (curculin-like)	+	
AT3G04720	ID	expressed protein (Barwin domain, defense protein)	+	
AT1G66970	LM	glycerophosphoryl diester phosphodiesterase (GDPDL1, GDPL3)	+	
AT1G55260	LM	non-specific lipid transfer protein (AtLTPg6)	+	
AT2G14610	M	PR1/Cys-rich secretory protein (SCP)	+	
AT1G55210	M	dirigent protein (AtDIR20)	+	
AT1G78680	P	peptidase C26 (peptidase family C26.003, MEROPS) (GGH2)	+	
AT5G08370	PAC	GH27 (α-galactosidase/melibiase)	+	
AT3G16850	PAC	GH28 (polygalacturonase)	+	
AT2G18660	PAC	expansin-like (AtEXR3)	+	
**AT5G38980**	UF	expressed protein	+	
AT3G28220	UF	expressed protein (MATH domain)	+	
AT5G48540	UF	expressed protein	+	
AT4G29240	ID	expressed protein (LRR domains)		+
AT4G12390	ID	plant invertase/pectin methylesterase inhibitor (PMEI-like)		+
AT1G49740	LM	expressed protein (phospholipase C domain)		+
AT5G03610	LM	lipase acylhydrolase (GDSL family)		+
AT3G08770	LM	non-specific lipid transfer protein (AtLTP1.6, AtLTP6)		+
AT3G14920	PAC	peptide-N4-(N-acetyl-β-glucosaminyl) asparagine amidase A		+
**AT5G38980**	UF	expressed protein		+
AT4G28100	UF	expressed protein		+
AT2G47010	UF	expressed protein		+

The statistical analysis of the data was performed with a sparse MB-PLS-DA and selected the best hit among each block. ^1^ AGI codes in bold mean that the protein and the corresponding transcripts were both differentially accumulated. ^2^ See legend to Table 2, for the meaning of the following abbreviations: ID, LM, OR, P, PAC, UF. ^3^ “+” corresponds to proteins or transcripts showing an increased level of accumulation. The data fold changes be read in Appendix A.

**Table 5 cells-09-02249-t005:** Synopsis of CWPs or CW transcripts differentially accumulated in floral stems of Hosp and Hern.

AGI Code ^1^	Functional Class ^2^	Putative Function	Proteins ^3^	Transcripts ^3^	Hosp ^3^	Hern ^3^
AT1G66970	LM	glycerophosphoryl diester phosphodiesterase (GDPDL1, GDPL3)	+		+	
AT1G27950	LM	non-specific lipid transfer protein (AtLTPg4, LTPG1)	+		+	
AT3G32980	OR	class III peroxidase (AtPrx32) (ATP16A)	+		+	
AT2G12480	P	Ser carboxypeptidase (AtSCPL43)	+		+	
AT3G02740	P	Asp protease (peptidase family A01.A23, MEROPS)	+		+	
AT4G31140	PAC	GH17 (β-1,3-glucosidase)	+		+	
AT5G60950	S	AtCOBL5	+		+	
**AT5G19240**	UF	expressed protein	+		+	
AT3G15356	ID	lectin (legume lectin domain)	+			+
AT3G16530	ID	lectin (legume lectin domain)	+			+
AT3G04720	ID	expressed protein (PR4, Barwin domain, defense protein)	+			+
AT2G26010	M	homologous to gamma thionin (defensin)	+			+
AT4G16260	PAC	GH17 (β-1,3-glucosidase)	+			+
AT1G29050	PAC	homologous to *A. thaliana* PMR5 (carbohydrate acylation)	+			+
AT1G78820	ID	lectin (curculin-like)		+	+	
AT2G42800	ID	expressed protein (LRR domains)		+	+	
AT2G17120	ID	expressed protein (LysM domain)		+	+	
AT5G03610	LM	lipase acylhydrolase (GDSL family)		+	+	
AT4G39640	M	gamma glutamyltranspeptidase (GGT2)		+	+	
AT4G02330	PAC	CE8 (AtPME41)		+	+	
AT5G09760	PAC	CE8 (AtPME51)		+	+	
AT1G78060	PAC	GH3 (β-xylosidase) (AtBXL7)		+	+	
AT5G42100	PAC	GH17 (β -1,3-glucosidase)		+	+	
AT4G34480	PAC	GH17 (β -1,3-glucosidase)		+	+	
AT4G01700	PAC	GH19		+	+	
AT3G02880	S	recepteur kinase (LRR III subfamily)		+	+	
**AT5G19240**	UF	expressed protein		+	+	
AT5G48540	UF	expressed protein		+	+	
AT3G44100	LM	expressed protein (MD-2-related lipid-recognition)domain		+		-
AT1G31550	LM	lipase acylhydrolase (GDSL family)		+		-
AT1G01300	P	Asp protease (peptidase family A01.A05, MEROPS)		+		-
AT3G61820	P	Asp protease (peptidase family A01.A13, MEROPS)		+		-
AT3G12700	P	Asp protease (peptidase family A01.A30, MEROPS)		+		-
AT3G52500	P	Asp protease (peptidase family A01.A47, MEROPS)		+		-
AT5G19740	P	peptidase M28 (peptidase family M28.A02, MEROPS)		+		-
AT2G22970	P	Ser carboxypeptidase (AtSCPL11)		+		-
AT3G02110	P	Ser carboxypeptidase (AtSCPL25)		+		-
AT5G23210	P	Ser carboxypeptidase (AtSCPL34)		+		-
AT1G28110	P	Ser carboxypeptidase (AtSCPL45)		+		-
AT5G51750	P	Ser protease (AtSBT1.3)		+		-
AT4G00230	P	Ser protease (AtSBT4.14, XSP1)		+		-
AT5G25980	PAC	GH1 (AtBGLU37)		+		-
AT1G70710	PAC	GH9 (AtCEL1)		+		-
AT4G02290	PAC	GH9 (endo-1,3(4)-β-glucanase)		+		-
AT4G23820	PAC	GH28 (polygalacturonase)		+		-
AT4G33220	PAC	CE8 (pectin methylesterase – PME) (AtPME44)		+		-
AT5G23870	PAC	CE13 (pectin acylesterase - PAE) (AtPAE9)		+		-
AT1G04680	PAC	PL1 (pectate lyase) (AtPLL26)		+		-

The statistical analysis of the data was performed with a sparse MB-PLS-DA and selected the best hit among each block. ^1^ AGI codes in bold mean that the protein and the corresponding transcripts were both differentially accumulated. ^2^ See legend to Table 2, for the meaning of the following abbreviations: ID, LM, M, OR, P, PAC, S, UF. ^3^ “+/−“correspond to proteins or transcripts showing an increased (+) or a decreased (–) level of accumulation. The fold changes can be read in Appendix A.

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
