# Peer review of "An Integrative Study Showing the Adaptation to Sub-Optimal Growth Conditions of Natural Populations of Arabidopsis thaliana: A Focus on Cell Wall Changes"

_cells, 2020, doi:10.3390/cells9102249_

Round 1

Reviewer 1 Report

I have reviewed the manuscript entitled “An integrative study showing the adaptation to sub-optimal growth conditions of natural populations of Arabidopsis thaliana: a focus on cell wall changes” (MS#936637) by Duruflé et al, which was submitted to Cells. This is an interesting manuscript because it pursues to unravel the mechanistic basis of adaptation in Arabidopsis along elevation/temperature gradients using natural populations. I agree with the authors when they state that this is in fact a neglected issue in the literature. Although I appreciate the work conducted in this study, I have some concerns that, in my opinion, deserve to be considered by the authors in order to gain further clarity and impact. Below I go through them in detail.

  1. The introduction tackles all the points of interest, but perhaps not in the best way. For example, I had to read twice the first sentence with a definition of local adaptation. It is not completely clear to me. Despite the confusing first sentence, the key of this introduction is in lines 53 and 54, where the authors stress their major point. However, after a sentence like that I would expect a development of the idea outlining why this is important. There may be several options to do that, but the authors need to explain why studying cell walls is relevant here for that particular purpose. The next two paragraphs after posing the problem deal with the study species and cell walls, but there is no logical connection among all parts. In other words, the authors can move away from standard procedures to bring out their real motivations and ideas behind this study. Finally, I strongly suggest posing working hypotheses to be tested in this study. Again, the authors have different options to do that because the results show that in some cases altitude matters more than genetic background for some traits and vice-versa for others, which must have implications to better understand the process of local adaptation.

  1. The style in which the manuscript is written can also be considered. For example, there is a massive abuse of the passive voice. If the authors combine the active and passive voice, the comprehension of the text will increase substantially. This comment applies throughout the entire text.

  1. L87-88. Although the altitude of Col in Poland is considered to be around 200 m, I am sure that today’s Arabidopsis from that region differ a lot from Col. This is because Col has been accumulating mutations for decades and it is no longer an ecotype, but a lab strain. I only had some trouble placing Col on an equal footing with regard to natural accessions and ecotypes.

  1. Overall, the materials and methods section requires some improvement. My major concern is that the reader has to read the literature cited to know how the authors undertook the work. I understand that all protocols have already been used and published, but the literature is useful to get into deeper detail if needed, but not for the basics. I suggest including a brief description of the major steps (those important to make sure that all quality standards were met) for all methodologies described in this manuscript. Please note that to review this manuscript in detail, I should read half a dozen of papers to know the details, which is not possible. The reader does not want that either.

  1. I also have concerns on the way the authors have exploited their data. The authors have a nice experimental design including four populations from low and high altitudes with different genetic backgrounds. This means that the authors can test the effects of such fixed factors on the variables generated in this study. Overall, I found a bit confusing whether authors used replicates of different individuals within populations as replicates at all times (I think it is the former). Whatever the case, the authors have the means to go beyond the current analyses to focus on how the two factors mentioned play a role. Reading the results and the discussion, one feels that the authors just describe the findings, when one would expect them to test hypotheses.

  1. As mentioned throughout the manuscript, phenotypic plasticity is important to understand the potential to perform well in different environments. However, the authors did not quantify phenotypic plasticity as such. See examples applicable to this study in Botto (PCE, 2015, doi:10.1111/pce.12481) and Méndez-Vigo et al. (PCE, 2016, doi:10.1111/pce.12608). I am sure that including specific phenotypic plasticity estimates will increase the potential to exploit the data, which will also contribute to gain clarity.

  1. Finally, I admit that the discussion is a bit hard to follow. I understand (and appreciate) the complexity of the results, which is an asset in this study. However, the conclusions of this study are not clear given the statements of the introduction. Again, structuring the introduction in a different way and posing hypotheses (and the precise way to test them) will also help provide structure to the discussion section, given the close link between them.

Other comments:

  1. I could not find how the authors obtained the genetic structure to detect clusters. A small graph next to Table 1 would be welcome.

  1. I understand that any genotype required vernalization to flower. In fact, they all seem to be early flowering genotypes. In Spain, genotypes above 800 show an obligate vernalization requirement to become reproductive.

  1. I would use different symbols for different temperatures and five colors only for genotypes in all figures. As it is, the authors need to use 10 colors, which is a bit too much. I would be consistent in all of them because codes with numbers overlap.

Xavier Picó

Doñana Biological Station, CSIC, Seville, Spain

Author Response

  1. The introduction tackles all the points of interest, but perhaps not in the best way. For example, I had to read twice the first sentence with a definition of local adaptation. It is not completely clear to me. Despite the confusing first sentence, the key of this introduction is in lines 53 and 54, where the authors stress their major point. However, after a sentence like that I would expect a development of the idea outlining why this is important. There may be several options to do that, but the authors need to explain why studying cell walls is relevant here for that particular purpose. The next two paragraphs after posing the problem deal with the study species and cell walls, but there is no logical connection among all parts. In other words, the authors can move away from standard procedures to bring out their real motivations and ideas behind this study. Finally, I strongly suggest posing working hypotheses to be tested in this study. Again, the authors have different options to do that because the results show that in some cases altitude matters more than genetic background for some traits and vice-versa for others, which must have implications to better understand the process of local adaptation.

The introduction has been reorganized to better explain the aim of the study and particularly to strengthen the importance of the modifications of the cell wall in response to environmental constraints.

  1. The style in which the manuscript is written can also be considered. For example, there is a massive abuse of the passive voice. If the authors combine the active and passive voice, the comprehension of the text will increase substantially. This comment applies throughout the entire text.

The text has been modified accordingly.

  1. L87-88. Although the altitude of Col in Poland is considered to be around 200 m, I am sure that today’s Arabidopsis from that region differ a lot from Col. This is because Col has been accumulating mutations for decades and it is no longer an ecotype, but a lab strain. I only had some trouble placing Col on an equal footing with regard to natural accessions and ecotypes.

We agree to consider Col as the “plant pet lab”, but, we have recently sequenced part of Col and the sequences obtained match at 100% with the Col sequenced in 2000. It could also be mentioned that at least one other study uses Col as a natural ecotype although it has been grown in laboratory conditions for many years (https://www.frontiersin.org/articles/10.3389/fpls.2016.01026/full).

In this revised version of the manuscript, we have indicated that Col was originally collected at an altitude of 200 m in the Material and methods section and in Table 1.

  1. Overall, the materials and methods section requires some improvement. My major concern is that the reader has to read the literature cited to know how the authors undertook the work. I understand that all protocols have already been used and published, but the literature is useful to get into deeper detail if needed, but not for the basics. I suggest including a brief description of the major steps (those important to make sure that all quality standards were met) for all methodologies described in this manuscript. Please note that to review this manuscript in detail, I should read half a dozen of papers to know the details, which is not possible. The reader does not want that either.

This work has required many techniques to collect all the omics data. Their detailed description would have been lengthy and would have probably lead to auto-plagiarism since we have already published all of them. However, we have added some information in the Material and Method section regarding the extraction of the cell wall proteins and the polysaccharides (§ 2.3, 2.4 and 2.5).

  1. I also have concerns on the way the authors have exploited their data. The authors have a nice experimental design including four populations from low and high altitudes with different genetic backgrounds. This means that the authors can test the effects of such fixed factors on the variables generated in this study. Overall, I found a bit confusing whether authors used replicates of different individuals within populations as replicates at all times (I think it is the former). Whatever the case, the authors have the means to go beyond the current analyses to focus on how the two factors mentioned play a role. Reading the results and the discussion, one feels that the authors just describe the findings, when one would expect them to test hypotheses.

We have clarified what we call “biological replicates” in the Material and Method section (§ 2.1). We have chosen one genotype per homogeneous population, and 20 plants of this genotype for each biological replicate for all the analyses. We are not studying intra-population variability.

Besides, our approach relies on exploratory rather than statistical testing methods, because we think that they are more adapted to deal with heterogeneous datasets. We agree that 2-factor ANOVA or linear modeling would reveal more precisely the potential effects of both factors on the numerical variables, but this would not allow unraveling multivariate relationships between variables inside one data set or between two data sets.

  1. As mentioned throughout the manuscript, phenotypic plasticity is important to understand the potential to perform well in different environments. However, the authors did not quantify phenotypic plasticity as such. See examples applicable to this study in Botto (PCE, 2015, doi:10.1111/pce.12481) and Méndez-Vigo et al. (PCE, 2016, doi:10.1111/pce.12608). I am sure that including specific phenotypic plasticity estimates will increase the potential to exploit the data, which will also contribute to gain clarity.

We are not sure that we understand the question. However, we believe that the statistical framework that we have developed using this integrative analysis is very effective in highlighting new behaviors at a multi-biological level. In addition, more parameters need to be analysed to quantify phenotypic plasticity, which was not the purpose of this study. Altogether, our study was not designed in this purpose, so it seems complicated to explore our data with this approach without any analysis bias.

  1. Finally, I admit that the discussion is a bit hard to follow. I understand (and appreciate) the complexity of the results, which is an asset in this study. However, the conclusions of this study are not clear given the statements of the introduction. Again, structuring the introduction in a different way and posing hypotheses (and the precise way to test them) will also help provide structure to the discussion section, given the close link between them.

We agree that the Discussion is complicated. We hope that the changes introduced in the manuscript will clarify the conclusions.

Other comments:

  1. I could not find how the authors obtained the genetic structure to detect clusters. A small graph next to Table 1 would be welcome.

More information about the genetic clusters is given in Table 1.In addition, we have modified the nomenclature of clusters (I, II and III instead of A, B and C) to fit with their description published in ref 38.

  1. I understand that any genotype required vernalization to flower. In fact, they all seem to be early flowering genotypes. In Spain, genotypes above 800 show an obligate vernalization requirement to become reproductive.

When we have characterized the 30 A. thaliana Pyrenean populations, we were surprised to see that only one of them (named Pont) required vernalization (see ref. 38). Then, none of the four populations chosen for this study requires vernalization to flower.

  1. I would use different symbols for different temperatures and five colors only for genotypes in all figures. As it is, the authors need to use 10 colors, which is a bit too much. I would be consistent in all of them because codes with numbers overlap.

We have tried to replace the lettering by symbols, but the figures were not more readable. Then, we propose to add ellipses to underline the populations which segregated and which are discussed in the text. Additional comments have been added in the text to facilitate the reading of the figures.

Reviewer 2 Report

The article” An integrative study showing the adaptation to 2 sub-optimal growth conditions of natural populations 3 of Arabidopsis thaliana: a focus on cell wall changes” by Duruflé et al have interesting subject matters and they have taken up a very significant issue in the field to understand the environmental effect on change in the biochemical composition of a plant cell wall. Although the problem is not completely innovative but improves our current understanding of the subject.  I have the following concern about the manuscript that needs to be taken care of.

  1. Minor issues are related to the overuse of abbreviations throughout the manuscript that makes it difficult to read and understand the manuscript.
  2. Mass spec data was used to predict the protein variability in isolated cell wall fraction and based on bioinformatics analysis identified proteins were predicted whether they belong to the cell wall or not. I would suggest to further verify them experimentally for their cell wall localization.
  3. The figures in the manuscripts are not very well presented and clear. The overlapping labeling is making them difficult to understand. I would suggest finding a better way to present these graphs. There is variability in font size. Figure legends should mention color coding used in the block diagram in fig 6, 7, etc.
  4. Line 403; “thanks” is not a suitable word to use. It does not appear scientific.
  5. Line 412-414, this line is confusing. Line 413 suggesting that the lowest correlation of 0.63 exist between phenomic and metabolomic in rosettes however figure 6A, the correlation between these two is 0.85. This line should be corrected.
  6. All tables (2, 3, 4, and 5 ) should have information about fold change in differentially expressed transcripts among different variables.
  7. One of the major concerns about this paper is the lack of physiological significance of changes in cell wall composition. They should be properly discussed and correlated.
  8. Line 647, discussed atltp3 mutant but no data has been presented in the manuscript. if this has been discussed based on the previous finding, the ref. should be included.

Author Response

  1. Minor issues are related to the overuse of abbreviations throughout the manuscript that makes it difficult to read and understand the manuscript.

Most of the abbreviations have been removed. A list of the remaining ones is now provided at the end of the text.

  1. Mass spec data was used to predict the protein variability in isolated cell wall fraction and based on bioinformatics analysis identified proteins were predicted whether they belong to the cell wall or not. I would suggest to further verify them experimentally for their cell wall localization.

We agree that checking the sub-cellular localization of a protein of interest experimentally is always better than a bioinformatics prediction. However, it would be a huge task for so many candidate proteins.

However, in addition to bioinformatics prediction of subcellular localization, we check our findings in two ways: (i) we look in the literature whether some localization is available for the identified protein families and (ii) we compare our results to those of previous cell wall proteomics studies using a database dedicated to cell wall proteomes (WallProtDB, www.polebio.lrsv.ups-tlse.fr/WallProtDB). This information has been added in Material and Methods (§ 2.8).

We also wish to say that we have already experimentally checked the sub-cellular localization of about 30 proteins predicted to be localized in the cell wall using the same bioinformatics pipeline. Until now we have only found one of these proteins located in the vacuole (unpublished work) and another one located in both the cell wall and chloroplasts (https://www.frontiersin.org/articles/10.3389/fpls.2017.00263/full).

  1. The figures in the manuscripts are not very well presented and clear. The overlapping labeling is making them difficult to understand. I would suggest finding a better way to present these graphs. There is variability in font size. Figure legends should mention color coding used in the block diagram in fig 6, 7, etc.

We have tried to replace the lettering by symbols, but the figures were not more readable. The overlapping labelling is a result itself because it means that the populations do not discriminate properly with the tested parameters. The labelling are only readable when the discrimination is significant. Then, we propose to add ellipses to focus on the major differences which are discussed in the text. Comments have also been added in the text to facilitate the reading of the figures.

Figures 6 and 7 have been simplified to increase their readability. Former panels B of both Figures 6 and 7 have been put in supplementary data.

  1. Line 403; “thanks” is not a suitable word to use. It does not appear scientific.

The text has been modified accordingly.

  1. Line 412-414, this line is confusing. Line 413 suggesting that the lowest correlation of 0.63 exist between phenomic and metabolomic in rosettes however figure 6A, the correlation between these two is 0.85. This line should be corrected.

This is not a mistake. Actually the text was referring to former Supplementary Fig. S5 (now Supplementary Fig. S6).

  1. All tables (2, 3, 4, and 5) should have information about fold change in differentially expressed transcripts among different variables.

In this work, we intend at taking benefit from the relationships inside each numerical data set and between data sets, rather than considering differentially accumulated transcripts /proteins per se. Displaying fold-change would not bring more information and would be complicated since they may be different for the different populations. We know that this is not a standard approach but we believe that integrative studies must favor this kind of exploratory approaches instead of relying on standard ones based on statistical testing.

However, for people interested in these values, we have added a reference to the Supplementary Tables where they are displayed in the legends to Tables 2-5.

  1. One of the major concerns about this paper is the lack of physiological significance of changes in cell wall composition. They should be properly discussed and correlated.

We have added some information about the role of cell walls in the plant response to environmental constraints in the Introduction. Then, we have already discussed the possible relationships between the observation of an increase in the cuticle hydrophobicity at the 15°C growth temperature and the identification of candidate genes/proteins possibly involved in lipid metabolism in the Discussion (§ 4.2, 2nd paragraph). We have now added comments about the observed modifications at the level of cell wall polysaccharides (§ 4.2, 1st paragraph).

  1. Line 647, discussed atltp3 mutant but no data has been presented in the manuscript. if this has been discussed based on the previous finding, the ref. should be included.

The reference has been added.

Reviewer 3 Report

This manuscript aims to analyze the effects of different environmental conditions on certain histological and molecular parameters of Arabidopsis, comparing different altitudes and temperatures. In particular, the authors intend to analyze the effects of conflicting environments on the cell wall at the polysaccharidic and protein level. The work is part of the research into the effects of climate change on plant growth and reproduction. The introduction is well written, it seems to me that it adequately introduces the problem. The justification for the use of Arabidopsis is appropriate, I think it would also be useful to motivate more the choice of the two temperatures analyzed (why 22 and 15 degrees?). Moreover, it is true that the cell wall is a very well-known structure; however, since it represents the subject of study, a few more words about the relationship between cell wall and abiotic stress could be useful to frame more the work.

The approach to work is multidisciplinary and based on bioinformatics, proteomic and even histological analyses. This produced a significant set of data.

The data of protein analyses is interesting, although from an initial observation it is not perceived exactly the impact and meaning; I understand that the authors focus more on protein clustering rather than on the biological meaning of the absence/presence of specific proteins. Why not enter into the substance of what the lower/higher amount or absence/presence of proteins may imply, especially in relation to the previous analysis of carbohydrates?

The data on more or less hydrophobic polysaccharides of Figure 2 is interesting and also very striking. Is there evidence of differently expressed proteins or genes that support this histological difference?

I wonder if, in addition to identifying putative genes involved in the different behavior of varieties, functions cannot also be assigned to these genes or otherwise try to frame these genes in a broad model. I see that the authors try to assign hypothetical functions to genes that are differently expressed but I do not perceive a global model, but rather individual instances. I understand that in such cases of large data sets, it is not trivial to sew all the data together.

I admit that the work contains very interesting data, even a long list of data, with the aim of finding gene correlations between the different varieties within their different behavior; however, the work does not provide precise indications of genes/proteins underlying these different behaviors. I understand that the work moves with the aim of defining groups of differences and similarities but from a purely biological point of view I cannot personally become attached to this type of study. I see this work more useful in perspective. For the time being, I give a fairly positive assessment, but only on the basis of the amount of data that the authors have produced.

Author Response

This manuscript aims to analyze the effects of different environmental conditions on certain histological and molecular parameters of Arabidopsis, comparing different altitudes and temperatures. In particular, the authors intend to analyze the effects of conflicting environments on the cell wall at the polysaccharidic and protein level. The work is part of the research into the effects of climate change on plant growth and reproduction. The introduction is well written, it seems to me that it adequately introduces the problem. The justification for the use of Arabidopsis is appropriate, I think it would also be useful to motivate more the choice of the two temperatures analyzed (why 22 and 15 degrees?). Moreover, it is true that the cell wall is a very well-known structure; however, since it represents the subject of study, a few more words about the relationship between cell wall and abiotic stress could be useful to frame more the work.

The choice of the two growth temperatures has been explained in the Introduction (last paragraph). In addition, we have added more information about the cell wall.

The approach to work is multidisciplinary and based on bioinformatics, proteomic and even histological analyses. This produced a significant set of data.

The data of protein analyses is interesting, although from an initial observation it is not perceived exactly the impact and meaning; I understand that the authors focus more on protein clustering rather than on the biological meaning of the absence/presence of specific proteins. Why not enter into the substance of what the lower/higher amount or absence/presence of proteins may imply, especially in relation to the previous analysis of carbohydrates?

The discussion has been extended to take into account this comment about cell wall polysacccharides (§ 4.2).

The data on more or less hydrophobic polysaccharides of Figure 2 is interesting and also very striking. Is there evidence of differently expressed proteins or genes that support this histological difference?

The differences observed by histological staining rather refer to hydrophobic compounds which could be either phenolic or lipidic compounds. This is discussed in §4.2.

I wonder if, in addition to identifying putative genes involved in the different behavior of varieties, functions cannot also be assigned to these genes or otherwise try to frame these genes in a broad model. I see that the authors try to assign hypothetical functions to genes that are differently expressed but I do not perceive a global model, but rather individual instances. I understand that in such cases of large data sets, it is not trivial to sew all the data together.

We believe that it is too early to propose a broad model integrating all these data. However, we have tried to connect different observations, the most conclusive one being the increase in the cuticle hydrophobicity (see the Discussion section). Another difficulty is that the precise biological function of all the identified candidates is not established as stressed in the Conclusions.

I admit that the work contains very interesting data, even a long list of data, with the aim of finding gene correlations between the different varieties within their different behavior; however, the work does not provide precise indications of genes/proteins underlying these different behaviors. I understand that the work moves with the aim of defining groups of differences and similarities but from a purely biological point of view I cannot personally become attached to this type of study. I see this work more useful in perspective. For the time being, I give a fairly positive assessment, but only on the basis of the amount of data that the authors have produced.

We agree with this comment. This work should be considered as a first step towards the understanding of features critical for the adaptation of plants to their environment. Of course, the genes/proteins which have been identified now need to be further investigated.

Round 2

Reviewer 1 Report

I have reviewed the revised version of the manuscript entitled “An integrative study showing the adaptation to sub-optimal growth conditions of natural populations of Arabidopsis thaliana: a focus on cell wall changes” (MS#936637.R1) by Duruflé et al, which was submitted to Cells. Overall, the manuscript has gained clarity, despite its inherent complexity. This is a nice example of the intricacy of local adaptation in plants, as well as of how the combination of approaches (phenomics, metabolomics, proteomics and transcriptomics) on a handful of selected natural populations can yield interesting results. With no doubt, this study can open novel research avenues to keep understanding the genetic basis of the process of local adaptation.

Xavier Picó

Doñana Biological Station, CSIC, Seville, Spain

Reviewer 2 Report

The authors have addressed most of my concerns raised in my previous comments. Now the manuscript is improved. Still, it will be interesting to see the validation of a representative predicted cell wall protein.  There are minor typos in manuscripts.